# Agreement Between the Gross Motor Ability Estimator-3 and the Reduced Gross Motor Function Measure-66 Based on Artificial Intelligence

**DOI:** 10.3390/jcm14134512

**Published:** 2025-06-25

**Authors:** Stefanie Steven, Carlotta Müller, Karoline Spiess, Christiane Bossier, Eckhard Schönau, Ibrahim Duran

**Affiliations:** 1Center of Prevention and Rehabilitation, Medical Faculty and University Hospital, University of Cologne, 50931 Cologne, Germany; stefanie.steven@unireha-koeln.de (S.S.); cmuel126@smail.uni-koeln.de (C.M.); karoline.spiess@unireha-koeln.de (K.S.); christiane.bossier@unireha-koeln.de (C.B.); eckhard.schoenau@unireha-koeln.de (E.S.); 2Department of Pediatrics, Medical Faculty and University Hospital, University of Cologne, 50931 Cologne, Germany

**Keywords:** cerebral palsy, gross motor function measure, artificial intelligence, children

## Abstract

**Background:** The reduced Gross Motor Function Measure-66 (rGMFM-66) has already demonstrated its validity compared to the standard GMFM-66 using the Gross Motor Ability Estimator-2 (GMAE-2). This study aimed to evaluate its validity using the updated Gross Motor Ability Estimator-3 (GMAE-3) and to compare agreement between GMFM-66v2 and GMFM-66v3. **Methods:** A retrospective analysis was conducted on 250 children with cerebral palsy (CP) enrolled in a rehabilitation program between 2015 and 2024. All GMFCS levels (I–V) were represented. The sample included 107 females and 143 males, with a mean age of 6.9 years (SD 3.4). Agreement between scoring methods was assessed using intraclass correlation coefficients (ICCs) and Bland–Altman analyses. **Results:** The rGMFM-66 showed excellent agreement with GMFM-66v3 (ICC = 0.994; 95% CI 0.992–0.996). Similar agreement was found between GMFM-66v2 andGMFM-66v3 (ICC = 0.994; 95% CI 0.991–0.996). Bland–Altman plots confirmed close agreement across all comparisons. The rGMFM-66 reduces administration time from 45 to 26 min, offering a 42% time saving in clinical use. **Conclusions:** The rGMFM-66 demonstrates very high agreement with GMFM-66v3 and appears to be a valid alternative. Its strong concordance supports its applicability in both clinical and research settings. Although agreement was high, minor differences between scoring methods indicate that results should be interpreted in light of the scoring algorithm applied.

## 1. Introduction

The Gross Motor Function Measurement-66 (GMFM-66) is a standardized assessment tool for measuring motor skills in children with cerebral palsy (CP). It enables a comprehensive evaluation of motor development, to assess therapy outcomes, and to track progress over time [1,2]. This instrument is the most widely used method worldwide for measuring motor skills in CP [3]. Motor skills are assessed in various positions, such as lying, sitting, and standing, as well as mobility skills such as crawling, walking, and running, using a Likert scale. The GMFM-66 is specifically developed and validated for children with CP [4]. It is a shortened version of the original GMFM-88, which includes 88 motor tasks designed to asses gross motor function in children with various motor impairments. The GMFM-66 was derived through Rasch analysis to improve clinical utility and measurement precision specifically for children with cerebral palsy [5]. It comprises 66 standardized motor tasks that cover five dimensions of gross motor function: lying and rolling, sitting, crawling and kneeling, standing, walking, running, and jumping. Each item is rated on a 4-point ordinal scale ranging from 0 (does not initiate) to 3 (completes the task fully). The rGMFM-66 is a shortened, computer-assisted version that uses an AI-based algorithm to select an individualized subset of approximately 34 items based on the child’s performance level. This tailored item selection reduces administration time while maintaining a high level of diagnostic accuracy [6,7].

Motor skills play a crucial role in daily independence, as they enable fundamental activities such as sitting, standing, and walking. Limitations in motor development can significantly impact social participation, autonomy, and overall quality of life [8]. A precise evaluation of these abilities is, therefore, crucial to designing individualized therapy plans and effectively promoting progress [9]. Furthermore, the GMFM-66 allows for the creation of reference curves that enable a comparison of a child’s motor development with that of their peers. This not only supports individual progress monitoring but also provides an evidence-based foundation for therapeutic decision-making [10].

To quantify motor function, the Gross Motor Ability Estimator-2 (GMAE-2) and its recently updated version, the Gross Motor Ability Estimator-3 (GMAE-3), are used. Both versions convert individual GMFM-66 item scores into a final motor function score (GMFM-66_v2_ and GMFM-66_v3_), ensuring a standardized evaluation process. Previous studies have demonstrated the compatibility of both estimators in calculating GMFM-66 values, confirming their comparability in clinical and research settings [1,6].

Despite its widespread use, the standard GMFM-66 assessment remains time consuming, which can pose challenges for both children and therapists [11]. To address these limitations, the reduced GMFM-66 (rGMFM-66) was developed with the goal of improving practicability and significantly reducing assessment time. By minimizing the testing burden, the rGMFM-66 allows for a more efficient allocation of resources and reduces the ethical strain on children undergoing repeated assessments [12,13]. A previous study successfully demonstrated the validity of the rGMFM-66 and documented its practical advantages. The validation process was conducted using the GMAE-2 as a reference standard [6]. The standard GMFM-66 typically requires around 45 min to administer, whereas the rGMFM-66 can be completed in approximately 26 min. This corresponds to a time reduction of about 42 percent, highlighting the practical benefit of the shortened version in everyday clinical practice [6]. This time demand can be particularly challenging for children with limited attention spans, fatigue, or behavioral difficulties. In addition, it places strain on therapists and reduces the feasibility of frequent reassessments in clinical rehabilitation settings [14].

Building on these findings, this study examined the agreement of the rGMFM-66 and the GMFM-66 using GMAE-3 as a reference in children and adolescents with CP. The agreement between the GMFM-66 scores calculated with GMAE-2 and GMAE-3 was also compared. It was hypothesized that the rGMFM-66, when evaluated using GMAE-3, would show excellent agreement with the GMFM-66. Additionally, we expected the GMFM-66 scores generated by GMAE-2 and GMAE-3 to be largely comparable.

## 2. Materials and Methods

### 2.1. Study Design and Study Population

We performed a retrospective analysis of data from children with CP enrolled in the rehabilitation program “Auf die Beine” at the UniReha GmbH, a Center for Prevention and Rehabilitation of the University of Cologne, between 2015 and 2024. The rehabilitation program “Auf die Beine” follows a structured protocol that includes the routine measurement of the GMFM-66 at the beginning of the rehabilitation. This tool is used to assess motor function and monitor progress throughout the rehabilitation process. The data were collected from the center’s patient registry. A description of the GMFM-66, including its structure and scoring system, is provided in the Section 1. Furthermore, additional inclusion criteria required that each child had data available for at least 65 items from the GMFM-66, ensuring an exact calculation of the GMFM-66 score. The sample was then narrowed down to include only children who were admitted for their first stay in the “Auf die Beine” rehabilitation program. These children were attending the center for the first time. No additional testing was conducted, and only pre-existing, routinely collected clinical data were analyzed. The overall cohort included 1123 children diagnosed with CP, aged 2 to 17 years.

In the literature, a median sample size of *n* = 42 is given for studies estimating the intraclass correlation coefficients (ICC) [15]. In addition, a sample size calculation was performed using the formula of Bonett for estimating ICC with pre-defined precision [16]. A sample size of 61 participants was calculated (with α = 0.05, number of rater = 2, expected value for ICC based on preliminary investigations = 0.99, width of the 95%CI = 0.01). All children included in the study were classified according to the Gross Motor Functioning Classification Scale (GMFCS), with levels ranging from I to V, representing a spectrum from mild to severe motor impairment. This classification was essential for ensure the homogeneity of the subgroup analysis in terms of motor function. The GMFCS is divided into five levels. Level I describes children with the least functional limitations. They are able to walk without restrictions at home, in school, outdoors, and in the community. Stairs can be managed without the use of a railing. However, activities requiring higher levels of coordination, such as running or jumping, may reveal mild limitations. Participation in physical activities or sports is generally possible, with only minor restrictions. Level II also includes children who are independently ambulatory in most everyday situations, but they may require mobility aids for longer distances, on uneven ground, or in crowded or stressful environments. Stairs can be climbed only with the help of a handrail, and a manual wheelchair may be used for longer distances. Participation in physical activities is possible but often limited and may require assistive devices. Children classified in Level III typically require hand-held mobility devices to walk indoors and may use wheeled mobility for longer distances. In Level IV, mobility is more severely limited; children usually rely on physical assistance or powered mobility even indoors. At Level V, children are completely dependent on caregivers for mobility and positioning and require extensive support for all activities of daily living. These levels reflect a continuum of motor function impairment and are essential for individualized rehabilitation planning [17]. We decided on a sample size of *n* = 50 for each GMFCS level. Thus, for the final analysis, the sample size was reduced to 250 children, who were randomly selected from the initial cohort. The random selection process aimed to provide a representative sample for children with CP and each GMFCS level (Figure 1).

### 2.2. Statistical Analyses

Statistical analysis was performed using R (version 4.4.1). To assess consistency between these measurement methods (the rGMFM-66 in comparison to the GMFM-66 calculated with GMAE-3, as well as GMFM-66 derived from GMAE-3 against GMFM-66 based on GMAE-2), the ICC was calculated using the irr package for R (version 0.84.1) [18]. The ICCs were calculated with the specifications of a two-way model, agreement, and single measurement. These analyses were performed both for the total sample and separately across GMFCS levels to identify potential variations in agreement across different levels of motor impairment. Additionally, a Bland–Altman analysis was conducted to quantify systematic differences between the methods.

## 3. Results

### 3.1. Study Population

The characteristics of the study population are presented in Table 1. The study included both female (*n* = 107) and male (*n* = 143) participants, with a total of 250 children represented across all GMFCS levels. The average age of the participants was 6.9 years (SD = 3.4). The mean weight was 21.2 kg, and the mean height was 114.0 cm, resulting in a calculated BMI of 16.3. The predominant CP subtype was bilateral spastic (68%), followed by mixed type (11.6%) and unilateral spastic (11.2%). In accordance with the SCPE classification system, bilateral spastic CP encompasses children formerly described as having diplegia or quadriplegia/tetraplegia, while unilateral spastic CP corresponds to what has previously been referred to as hemiplegia [19].

Statistical comparison across GMFCS levels showed significant differences in height and weight, with lower values observed in higher GMFCS levels (Table 1). Other variables, such as age and sex distribution, showed no significant group differences.

### 3.2. Comparison of GMFM-66_v3_ Versus rGMFM-66 and GMFM-66_v3_ Versus GMFM-66_v2_

The ICC for the comparison between the rGMFM-66 and GMFM-66_v3_ demonstrated a very high agreement of ICC = 0.994 (95%CI 0.992–0.996), which was very similar to the ICC for the comparison between GMFM-66_v2_ and GMFM-66_v3_ (0.994; 95%CI 0.991–0.996). Based on the classification proposed by Koo and Li (2016), both ICC values fall within the range of excellent reliability (ICC ≥ 0.90), confirming the strong consistency between scoring methods [20]. A similarly high level of agreement was observed in the calculations for individual GMFCS levels. The detailed results are presented in Table 2. In addition, a Pearson correlation plot visually illustrates the high agreement between the rGMFM-66 and GMFM-66v3 scores (Figure 2).

In addition, a Pearson correlation plot visually illustrates the high agreement between the rGMFM-66 and GMFM-66_v3_ scores (Figure 2). Diagram A shows the correlation between the rGMFM-66 and the GMFM-66, calculated using GMAE-3. Diagram B shows the correlation between the GMFM-66v_2_ and the GMFM-66v_3_. The Pearson correlation coefficient (*r* = 0.995, *p* < 0.001) also confirms this high correlation and underlines the strong agreement between the two assessment methods. Given the very high correlation between versions, these findings suggest that both scoring methods yield comparable results. However, small differences may still impact longitudinal tracking. However, this statistical agreement does not imply full interchangeability.

The Bland–Altman analysis confirms the similarly good agreement of GMFM-66_v3_ with rGMFM-66 and GMFM-66_v3_ with GMFM-66_v2_ (Figure 3). The mean difference between rGMFM-66 and GMFM-66_v3_ was –0.3 points, indicating a negligible negative bias. In comparison, the mean difference between GMFM-66_v2_ and GMFM-66_v3_ was close to 0, suggesting no systematic bias between these scoring versions.

## 4. Discussion

The present study demonstrates a very high agreement between the GMFM-66v3 and rGMFM-66 (ICC = 0.994; 95%CI 0.992; 0.996; Table 1; Figure 2). This finding supports the validity of the rGMFM-66 as an alternative to the conventional GMFM-66, also when using the recently updated calculating software (GMAE-3). Additionally, the high agreement between GMFM-66v3 and GMFM-66v2 (ICC = 0.994; 95%CI 0.991; 0.996) indicates that the updated algorithm does not introduce significant deviations from its previous version.

Previous studies by Duran et al. (2020) and Schafmeyer et al. (2023) validated the rGMFM-66 using the GMFM-66v2 as a reference and demonstrated its high validity (ICC 0.993; 95%CI 0.993; 0.995) [21,22]. Our results extend this evidence by confirming comparable validity with the GMFM-66v3. This reinforces the rGMFM-66’s applicability in both clinical and research settings, ensuring that it can be used alongside updated GMFM-66 calculation methods. In addition to confirming the validity, recent prospective findings reported in Steven et al. (2025) [6] provide robust evidence for the time-saving potential of the rGMFM-66. This study showed a reduction in assessment duration by approximately 42% compared with the standard GMFM-66, without compromising diagnostic accuracy. These results highlight that rGMFM-66 not only offers a statistically valid alternative but also a practically efficient tool for routine clinical use. This efficiency gain is particularly relevant in pediatric rehabilitation settings, where time constraints and child compliance are critical factors. The combination of strong agreement and reduced testing burden supports broader implementation of the rGMFM-66 in daily clinical workflows and longitudinal follow-up [6].

GMFM-66v2 is calculated using the GMAE-2 software, while GMFM-66_v3_ uses the updated GMAE-3 algorithm. GMAE-3 incorporates revised calibration data based on a larger and more diverse reference sample, thereby improving assessment precision and standard error estimation. While both estimation methods produce comparable results, the algorithmic improvements in GMAE-3 aim to improve robustness and accuracy across diverse clinical populations. This update reflects efforts to enhance the measurement stability and generalizability of the GMFM-66 score in an international context. Despite these changes, our results confirm that the transition to GMAE-3 does not significantly alter score results and supports continuity and comparability with previous applications of the GMFM-66 [1].

The agreement remained consistently high across all GMFCS levels, with minor differences observed at the lower GMFCS levels. For instance, an ICC of 0.945 (95%CI 0.893–0.971) was found for GMFCS level I, while an ICC of 0.983 (95%CI 0.970–0.990) was observed for level V. These differences seemed to be non-significant, since the 95% confidence intervals overlapped. In addition, all GMFCS levels show lower ICC than in the population as a whole. This difference is due to the method, as the ICC decreases when the variance within the group decreases [20].

Similar findings were also observed in the comparison of GMFM-66_v3_ versus GMFM-66_v2_ (Table 2). In contrast to these findings, Pierce et al. [1], who analyzed the agreement of GMFM-66_v2_ and GMFM-66_v3_ in n = 53 children with CP (aged 5–53 months), reported lower ICC in children with GMFM-66 < 25 (ICC 0.879; 95%CI 0.628–0.965) than in the whole study population (ICC 0.997; 95%CI 0.994–0.998). This difference could be due to static fluctuations with a small number of test subjects (n = 53 versus n = 250), or it may be due to the above-mentioned method-related reduction of the ICC with low variance in the group. Unfortunately, in the work by Pierce et al. [1], the ICC was not determined as GMFCS level dependent. The ICC (GMFM-66_v3_ versus GMFM-66_v2_) for the whole population in Pierce et al. [1] was very similar to that in our population.

The Bland–Altman analysis thus showed a similar agreement of the GMFM-66_v3_ with rGMFM-66 and of the GMFM-66_v3_ with the GMFM-66_v2_ (Figure 3). Since the upper and lower limits of agreement of the Bland–Altman diagram are approximately 4 points, the methods do not appear to be interchangeable when assessing an individual progression (in one patient). This means that if the development of a subject over time should be evaluated, it makes sense to use the same method (rGMFM-66, GMFM-66_v2_ or GMFM-66_v3_) at both measurement times. A comparison of groups of subjects whose motor skills were measured using different methods appears to be possible regardless of the method due to the high ICC values and very low systematic bias in the Bland–Altman diagram (mean difference of agreement is very close to zero, Figure 3).

The high correlation between the rGMFM-66 and the GMFM-66_v3_ has important clinical implications. The rGMFM-66 allows for a 50% reduction in assessment time without compromising diagnostic accuracy, as previously demonstrated by Steven et al. (2025) [6]. This highlights the flexibility and efficiency of rGMFM-66, making it a valuable tool in clinical practice by reducing the time burden for both clinicians and patients while maintaining diagnostic accuracy. Its applicability in various clinical settings allows for the efficient monitoring of motor function, particularly in rehabilitation programs where frequent assessments are necessary to track progress and adjust interventions accordingly [23].

The effectiveness and initial validity of rGMFM-66 have been demonstrated using the Gross Motor Ability Estimator-2 (GMAE-2), showing strong agreement with the standard GMFM-66_v3_. The present study expands on these findings by showing that the choice of scoring algorithm, whether GMAE-2 or the updated GMAE-3, has no substantial impact on the results, confirming the consistency and robustness of rGMFM-66 scoring across different estimation tools.

GMFM-66_v2_ is calculated using the GMAE-2 software, which has been widely used internationally since its introduction in the early 2000s [24]. The newer GMFM_v3_ uses the updated GMAE-3 algorithm, introduced in the early 2020s, which incorporates revised calibration data based on a larger and more diverse reference sample of children with CP. This update aimed to improve the accuracy of item difficulty estimates, reduce scoring variance, and enhance robustness in heterogeneous clinical populations. While both estimators produce highly comparable scores, GMAE-3 slightly refines standard error estimations and provides updated percentile references. In clinical practice and research, GMAE-2 has been the standard for many years and is still commonly used, particularly in institutions where the software transition is pending. However, the adoption of GMAE-3 is increasing, particularly in academic and international research contexts due to its methodological advancements [1]. The present study demonstrates that rGMFM-66 maintains excellent agreement with both versions, reinforcing its reliability and applicability regardless of the scoring version used. From a clinical standpoint, this is especially relevant for longitudinal monitoring and cross-site data comparison. It further supports the flexibility of rGMFM-66 and highlights its potential for broader implementation in modern rehabilitation settings.

Future research should build upon these findings by assessing additional measurement properties in accordance with the COSMIN guidelines. Recommended directions include the evaluation of responsiveness, test–retest reliability, and construct validity through hypothesis testing. Moreover, longitudinal studies across diverse clinical contexts may further support the instrument’s sensitivity to change and its role in long-term functional monitoring.

This is consistent with findings from Wang et al. (2006), who observed that the shortened GMFM-66 version showed better alignment with therapist assessments of motor improvement compared to the original GMFM-88 [25]. Similarly, Russel et al. (2000) confirmed the GMFM-66’s excellent validity and test-retest reliability, establishing it as an alternative to the full GMFM-88 item version [5]. In younger children under 3 years of age, Wei et al. (2006) found strong reliability and concurrent validity for the GMFM-66, supporting its use even in early development stages [26]. Further efficiency improvements were demonstrated by Russel et al. (2010) with the GMFM-66 Item Set (GMFM-66 IS), which applies an algorithm to select a reduced set of items while preserving score accuracy [12].

More recent research has focused on digital and retrospective adaptations. Duran et al. (2022) [7] developed rGMFM-66, a computer-based model that accurately predicts GMFM-66 scores using a reduced number of items, optimizing clinical feasibility [7]. The consistency of score calculation using the GMFM Estimator software version 2 and version 3 was further examined by Pierce et al. (2024) [1], who found high agreement between the outputs of version 2 (GMAE-2) and version 3 (GMAE-3), supporting score comparability across platforms [1]. A number of studies have previously examined the validity, reliability, and feasibility of shortened GMFM versions and associated estimator software. This section synthesizes these findings and places them in direct relation to the results of the present study. Table 3 offers a structured comparison of test versions, target populations, and key psychometric outcomes reported in the literature.

The ICC results presented in Table 3 demonstrate that the present study achieves levels of agreement that are fully consistent with those reported in previous validation studies. In the overall sample, agreement reached values comparable to the highest ICCs in the literature. Across individual GMFCS levels, the rGMFM-66 and both estimator versions (GMAE-2 and GMAE-3) showed robust and largely equivalent performance, further supporting the methodological soundness of the approach and the reliability of the updated scoring algorithm. To further contextualize these findings, the GMFM-66_v3_ scores were also compared with those obtained using the GMFM-66_v2_ estimator in the same dataset, resulting in nearly identical ICCs. These data suggest that the GMAE_v3_ of the GMFM-66 software does not introduce systematic changes in score output, and that rGMFM-66 scores calculated using GMAE-2 remain valid and comparable.

Particularly in GMFCS levels III to V, ICCs exceeded 0.95 in both comparisons, indicating highly consistent measurement even in children with more severe motor impairments. Slightly lower ICCs in levels I and II may reflect increased variability in motor performance and measurement sensitivity in higher-functioning children. These differences warrant further investigation in future studies, ideally with larger subsamples per GMFCS level, to determine whether they represent systematic measurement variation or sample-specific effects.

These findings are particularly relevant for clinical practice, as a shortened test version reduces the administrative burden for clinicians while simultaneously minimizing the strain on the assessed children [27]. This could further enhance the acceptance of rGMFM-66 in routine diagnostics.

Alongside the important findings, certain limitations must be addressed. As the study is based on existing clinical routine data, potential biases cannot be ruled out, and prospective studies are needed for further validation to confirm the generalizability of the present findings. Although the scoring methods yield highly comparable results, slight systematic differences—depending on the algorithm used—may influence interpretation in longitudinal assessments [20,25,28]. Therefore, to ensure consistent monitoring over time, we recommend maintaining the same scoring version throughout follow-up. This is particularly important in clinical settings where treatment progress is evaluated based on changes in GMFM-66 scores [5,13,25].

Additionally, while the sample size (*n* = 250) appears representative, subgroup analyses for individual GMFCS levels may have lower statistical power. In particular, comparisons of measurement agreement across subgroups should be interpreted with caution.

Another limitation is that rGMFM-66’s shortened nature might reduce sensitivity in detecting small motor function improvements in specific clinical contexts. This raises the question of whether it is equally effective in tracking gradual progress over time compared to the full GMFM-66. Upcoming studies should investigate whether the rGMFM-66 can reliably monitor long-term motor development and assess its applicability in younger children. Moreover, further research could explore its potential integration into digital health solutions, leveraging automated analysis for more efficient clinical decision-making.

Future studies should also prospectively investigate whether rGMFM-66 can be reliably used for long-term monitoring of motor development.

## 5. Conclusions

This study demonstrated an excellent agreement between rGMFM-66 and GMFM-66_v3_ (ICC = 0.994; 95% CI 0.992–0.996), supporting the rGMFM-66 as a valid and efficient alternative to the standard version. These findings confirm the high agreement between GMFM-66_v3_ and rGMFM-66, reinforcing its validity as an efficient alternative for motor function assessment in children with CP. In addition, a high agreement between the GMFM-66_v3_ and GMFM-66_v2_ was observed.

Despite the high statistical agreement between the different scoring methods, minor systematic differences were identified. These findings suggest that the scores may not be entirely interchangeable and should be interpreted within the context of the specific scoring algorithm used. These results underscore the clinical value of the rGMFM-66 in reducing assessment burden without compromising diagnostic precision, and they highlight the importance of standardized applications of scoring tools in pediatric rehabilitation. Consistency in the choice of method is therefore recommended, particularly in longitudinal monitoring and research settings.

Future research should explore the application of the rGMFM-66 in combination with GMAE-3, as well as its use in larger and more diverse populations. In particular, studies focusing on score variability in higher-functioning children (GMFCS I-II) and longitudinal follow-up will be important to refine the tool’s utility in clinical and research settings.

## Figures and Tables

**Figure 1 jcm-14-04512-f001:**
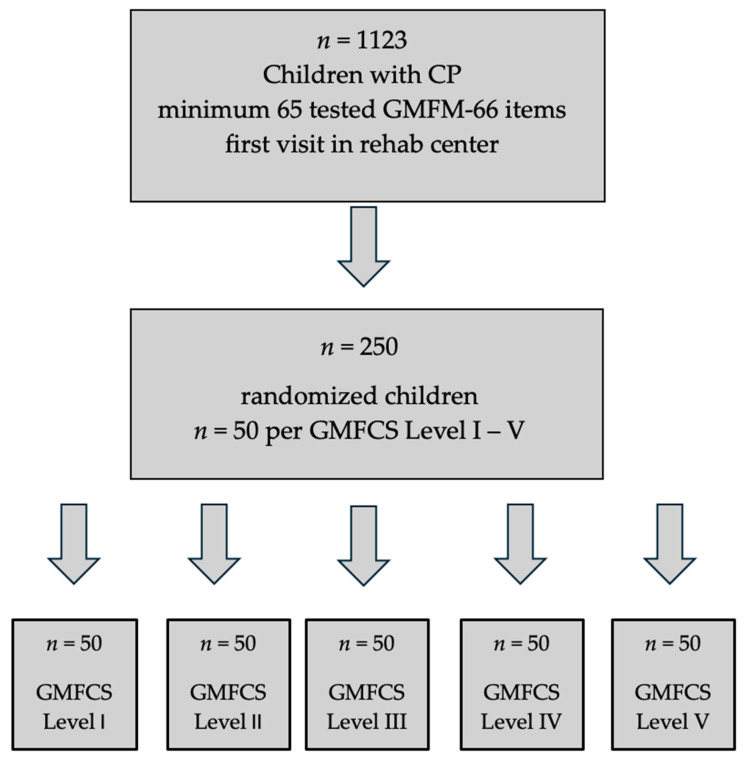
Flowchart illustrating participant selection and stratification. From a total of 1123 children with cerebral palsy (CP) assessed using at least 65 GMFM-66 items at their first visit to the rehabilitation center, 250 were randomly selected. Fifty children were allocated to each level of the Gross Motor Function Classification System (GMFCS I–V).

**Figure 2 jcm-14-04512-f002:**
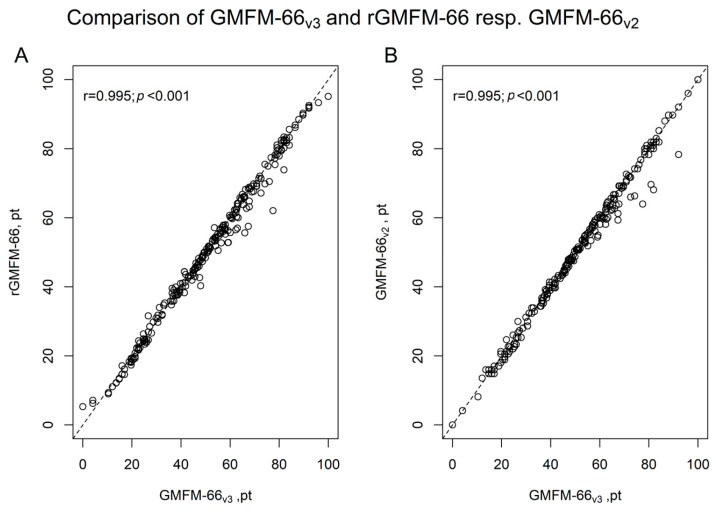
Correlation between GMFM-66_v3_ and (**A**) rGMFM-66, and (**B**) GMFM-66_v2_. Scatterplots showing the strong linear relationship between GMFM-66_v3_ and both the rGMFM-66 and GMFM-66_v2_. Pearson correlation coefficients indicate a near-perfect correlation in both comparisons (r = 0.995, *p* < 0.001).

**Figure 3 jcm-14-04512-f003:**
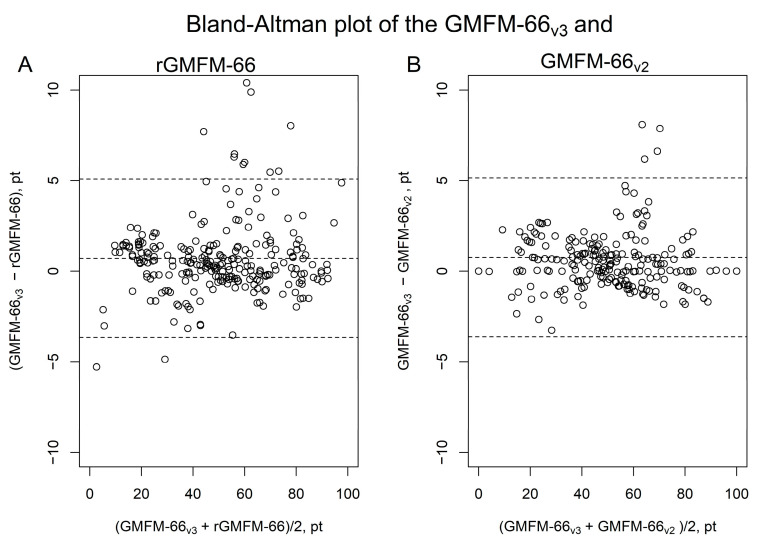
Bland-Altman plots comparing GMFM-66_v3_ with (**A**) rGMFM-66 and (**B**) GMFM-66_v2_**.** The plots illustrate the agreement between GMFM-66_v3_ and two alternative versions. The *y*-axis shows the difference between the scores, and the *x*-axis shows the mean of the two scores. Dashed lines represent the mean difference and the 95% limits of agreement.

**Table 1 jcm-14-04512-t001:** Baseline characteristics of the study population stratified by GMFCS levels I–V (*n* = 250). Values are present as mean (SD) or count (%). Superscript letters indicate statistically significant group differences (*p* < 0.005).

	GMFCS Level					
	I–V (*n* = 250)	I (*n* = 50)	II (*n* = 50)	III (*n* = 50)	IV (*n* = 50)	V (*n* = 50)
female, n	107 (42.8)	24 (44.0)	24 (42.0)	17 (34.0)	25 (50.0)	17 (34.0)
Age, years	6.9 (3.4)	7.6 (3.6)	7.4 (3.9)	6.6 (3.3)	6.6 (3.2)	6.3 (2.7)
Height, cm	114.0 (19.5)	122.3 (23.4) ^a^	119.4 (23.3)	110.3 (16.1)	110.4 (15.9)	107.4 (12.1) ^a^
Weight, kg	21.2 (11.4)	25.6 (12.2) ^b^	24.4 (14.3) ^c^	20.3 (9.5)	19.1 (11.2) ^b^	16.5 (5.8) ^b,c^
CP subtype, %						
Spastic bilateral	68.0	56.0	74.0	82.0	74.0	54.0
Spastic unilateral	11.2	38.0	12.0	4.0	2.0	0
Dyskinetic	7.2	0	0	8.0	8.0	20.0
Ataxic	2.0	4.0	4.0	2.0	0	0
Mixed type	11.6	2.0	10.0	4.0	16.0	26.0

^a^ Children with GMFCS I were significantly taller than children with GMFCS V (*p* = 0.016). ^b^ Children with GMFCS I were significantly heavier than children with GMFCS IV and V (*p* = 0.005, *p* = 0.001). ^c^ Children with GMFCS II were significantly heavier than children with GMFCS IV (*p* = 0.007).

**Table 2 jcm-14-04512-t002:** Intraclass correlation coefficients (ICC) between GMFM-66 version 3 and rGMFM-66, and between GMFM-66 version 3 and GMFM-66 version 2, stratified by GMFCS level (*n* = 250).

	GMFM-66_vs3_ Versus	
	rGMFM-66	GMFM-66_v2_
GMFCS Level	ICC	ICC
I–V (*n* = 250)	0.994 (0.992; 0.996)	0.994 (0.991; 0.996)
I (*n* = 50)	0.945 (0.893; 0.971)	0.920 (0.855; 0.955)
II (*n* = 50)	0.955 (0.897; 0.978)	0.967 (0.931; 0.983)
III (*n* = 50)	0.958 (0.926; 0.976)	0.987 (0.976; 0.993)
IV (*n* = 50)	0.986 (0.976; 0.992)	0.989 (0.979; 0.994)
V (*n* = 50)	0.983 (0.970; 0.990)	0.982 (0.961; 0.991)

The results are presented as mean (95% confidence interval).

**Table 3 jcm-14-04512-t003:** Chronological overview of key studies evaluating shortened versions of the GMFM and related estimator or software comparisons, including ICC values where applicable.

Study	Test Version/Focus	Population/Design	Key Findings
Wei et al. (2006) [26]	GMFM-66 in <3-year-olds	Children < 3 y with CP	ICC = 0.966–0.978; concurrent validity r = 0.985
Russell et al. (2010) [12]	GMFM-66 Item Set (GMFM-66 IS)	Validation sample (varied ages)	Algorithm-based item selection; ICC = 0.994 (single test); 0.92 (retest)
Duran et al. (2022) [7]	rGMFM-66	Retrospective sample with CP	Accurate prediction of GMFM-66; ICC = 0.997 (95% CI: 0.996–0.997)
Pierce et al. (2024) [1]	GMAE-2 vs. GMAE-3 (software only)	Software output comparison	High agreement between versions; confirms scoring equivalence
Steven et al. (2025) [6]	rGMFM-66	Children with CP (prospective sample)	Validated rGMFM-66; ICC = 0.970 (95% CI: 0.942–0.983); agreement with GMAE-2
Current study	rGMFM vs. GMFM-66_v3_	Children with CP (retrsospective sample)	Validated rGMFM66 with GMFM-66_v3_; ICC = 0.994 (95% CI: 0.992–0.996)

## Data Availability

The data supporting the findings of this study are not publicly available due to ethical reasons related to participant confidentiality.

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
