# Peer review of "Agreement Between the Gross Motor Ability Estimator-3 and the Reduced Gross Motor Function Measure-66 Based on Artificial Intelligence"

_jcm, 2025, doi:10.3390/jcm14134512_

Round 1
Reviewer 1 Report
Comments and Suggestions for Authors
The paper is interesting, but the presentation of the entire manuscript must be revised in accordance with the template.
1. The entire manuscript does not comply with the template, even the font style has been changed;
2. Due to the large line spacing, this paper de facto has only 6 pages and therefore cannot be categorized as an "Article", and the publication type should be changed accordingly; otherwise, the paper should be supplemented with a new section – literature review and analysis of findings, with a later review in the Discussion;
3. Line 46: "Previous studies have demonstrated..." Which studies? Provide appropriate references;
4. Line 57: "A previous study successfully..." Which studies? Provide appropriate references;
5. The data are well described independently of Fig. 1, above, so Fig. 1 can be removed due to data repetition;
6. Conclusion is too short and lacks directions for further research; therefore lines (225-230) should be moved to the Conclusion;
7. References in the Reference List also do not comply with the template;
Author Response
Please see the attachment.
Best Regards
Stefanie Steven

Reviewer 2 Report
Comments and Suggestions for Authors
The manuscript addresses an important and timely topic in pediatric rehabilitation: validating the shortened GMFM-66 (rGMFM-66) using the newer GMAE-3. The methodology is robust, the results are well-analyzed, and the conclusions are well-supported.
- line 14-16: clarify "its"
- line 21: Consider stating the exact ICC and confidence intervals here as was done in the previous sentence
- line 32: Verb tense shift
- line 45: The term “computer-based scoring software” might be too vague
- line 53: Consider adding quantitative estimate (e.g., % reduction)
- table 1: Table 1 is informative but lacks statistical comparisons between GMFCS levels
- line 121: consider explicitly noting that both comparisons fall within excellent reliability standards (ICC > 0.9)
- line 133: The paragraph is repeated (see lines 133–137 and again 139–143).
- line 152: Bland-Altman interpretation is adequate, but again, quantify the bias
- line 221: Suggest future studies to include responsiveness indices
- Abstract and conclusion should clarify limits of inter-method interchangeability
Author Response

(The authors gave the same response as above.)

Reviewer 3 Report
Comments and Suggestions for Authors
The study that is reviewed aimed to evaluate its validity using the updated Gross Motor Ability
Estimator-3 (GMAE-3) and to compare agreement between GMFM-66v2 and
GMFM-66v3.
Good topic that may add to literature. Here are recommendations to work on to improve your manuscript.
Introduction:
It was mentioned that standard GMFM-66 assessment remains time-consuming,
which can pose challenges for both children and therapists.you need to clearly identify the excat time it take and clarify what were these challnges .
Also need to write the time of the rGMFM66
Also need more clarification about components of both GMFM66 and rGMFM66
Need to show significance of the study and what it may addto literature. Also need to add hypothesis.
Methods : More details are needed regarfing study design and tools .time of the study more clarification of inclusion and exclusion criteria.
Need more clarification about GMFCS levels so that any reader can understand
Also need to add the measurements of GMFM-66
Results:
Why you just included females,??
In table1 cp classification should be corrected: spastic quadriplegia and tertaplegia
Hemiplegia
Results: you found agrrement between both
So in discussion you need to clarify what's the difference between both and what was the exact update .need to expand your discussion showing difference between all these versions what was excatly the update in each version how long it's being used and which one is most commmonly used in the past and nowadays international also need to show practical implications of your findings.
Conclusion should be more exclusive of paper findings.
Intext citation need to be corrected follow mdpi style the same ref in ref list should be rewritten using ACS , MDPI Style .
Author Response

(The authors gave the same response as above.)

Round 2
Reviewer 3 Report
Comments and Suggestions for Authors
All recommendations were done